# Advance Progress on Luminescent Sensing of Nitroaromatics by Crystalline Lanthanide–Organic Complexes

**DOI:** 10.3390/molecules28114481

**Published:** 2023-06-01

**Authors:** Yixia Ren, Zhihu Ma, Ting Gao, Yucang Liang

**Affiliations:** 1Laboratory of New Energy and New Function Materials, Shaanxi Key Laboratory of Chemical Reaction Engineering, College of Chemistry and Chemical Engineering, Yan’an University, Yan’an 716000, China; mazhihu123@163.com (Z.M.); 13636869137@163.com (T.G.); 2Institut für Anorganische Chemie, Eberhard Karls Universität Tübingen, Auf der Morgenstelle 18, 72076 Tübingen, Germany

**Keywords:** lanthanide–organic complex, luminescent material, luminescence sensing, nitroaromatics, mechanism

## Abstract

Water environment pollution is becoming an increasingly serious issue due to industrial pollutants with the rapid development of modern industry. Among many pollutants, the toxic and explosive nitroaromatics are used extensively in the chemical industry, resulting in environmental pollution of soil and groundwater. Therefore, the detection of nitroaromatics is of great significance to environmental monitoring, citizen life and homeland security. Lanthanide–organic complexes with controllable structural features and excellent optical performance have been rationally designed and successfully prepared and used as lanthanide-based sensors for the detection of nitroaromatics. This review will focus on crystalline luminescent lanthanide–organic sensing materials with different dimensional structures, including the 0D discrete structure, 1D and 2D coordination polymers and the 3D framework. Large numbers of studies have shown that several nitroaromatics could be detected by crystalline lanthanide–organic-complex-based sensors, for instance, nitrobenzene (NB), nitrophenol (4-NP or 2-NP), trinitrophenol (TNP) and so on. The various fluorescence detection mechanisms were summarized and sorted out in the review, which might help researchers or readers to comprehensively understand the mechanism of the fluorescence detection of nitroaromatics and provide a theoretical basis for the rational design of new crystalline lanthanide–organic complex-based sensors.

## 1. Introduction

Following the rapid development of modern industry, domestic wastewater, agricultural wastewater and pharmaceutical wastewater contain more and more pollutants, such as metal cations, anions, antibiotics, pesticides, nitro aromatic compounds and so on, which are making water environmental pollution more and more serious [1,2,3]. Among these pollutants, the toxic and explosive nitroaromatic compounds are often used as the starting materials for the preparation of dyes and explosives in the chemical industry. It should be noted that highly toxic nitroaromatics are extremely difficult to be degraded due to their structural and chemical stability. For example, nitrobenzene (NB) is the simplest and smallest molecule in nitroaromatics, which is often used as the intermediate and raw materials for the productions of dyes, fragrances, explosives and other aniline products in the organic synthesis industry. Inhalation of toxic NB vapor or skin contamination can cause acute poisoning, headache, nausea, vomiting and so on [4,5,6]. Furthermore, 4-Nitrophenol (4-NP) is mainly used as an intermediate in pesticide, pharmaceuticals, dyes and other fine chemicals [7,8]. In addition, 2,4,6-trinitrophenol (TNP), as the main component of explosives and gunpowder, is commonly used in the manufacture of dyes, preservatives and medicines. Wastewater from these industries needs to be monitored and treated to prevent environmental pollution [9,10]. Especially, high physiologically toxic nitroaromatics are harmful to (ground)water, soil resources and human health when released into the environment. Therefore, the detection of nitroaromatic compounds is of great significance to environmental monitoring, citizen safety and homeland security [11,12,13,14,15]. At present, the analytical methods for detecting harmful substances in the environment mainly include atomic absorption spectroscopy [16], mass spectrometry [17], optical gas chromatography-mass spectrometry [18], gas chromatography-electron capture detection [19], surface-enhanced Raman spectroscopy [20,21], X-ray imaging [21], thermal neutron analysis [21], electrochemical procedures [20] and ion mobility spectroscopy (IMS) [20,21] and molecularly imprinted polymers (MIPs) [22]. Note that fluorescence detection is a simple, convenient, highly sensitive method for monitoring pollutants in wastewater [23,24,25] and vapor phases [26,27,28], and even in commercial honey samples [29]; no special instructions are required compared with other methods, while rapid and real-time detection is provided, and fluorescence detection has demonstrated good sensitivity and selectivity.

Lanthanide ions have high coordination numbers and poor stereochemical preferences, sharp characteristic emission in the visible and near-infrared (NIR) ranges, large antenna-induced offsets and long lifetimes. Lanthanide–organic complexes have been widely investigated and applied in catalysis, adsorption, detection, luminescence, magnetism, separation, and so on [30,31,32]. In particular, lanthanide–organic complexes have the advantages of good luminescence performance, long emission lifetime, large Stokes shift and high quantum yield, and have broad application prospects in the field of fluorescence sensing [33,34]. It is well known that lanthanide complexes have a unique luminescence process, and the interaction between Ln^3+^ and an organic ligand coordinated to the Ln center with a fluorescence-sensing analyte can affect their emission intensity. Due to their unique properties, high coordination numbers and the flexible coordination modes of lanthanide metal ions, lanthanide–organic complexes possess diverse structures and excellent properties, in which the luminescence performance mainly stems from the metal-centered luminescence generated by absorbing radiative excitation energy from the excited state. The lanthanide ions regularly fill electrons in the 4f orbital, and the electron configuration is [Xe]4f_n_ (*n* = 0–14). The electronic structure generates well-defined energy levels, leading to interesting luminescent properties. The 4f orbital is insensitive to the outer electrons because it’s well shielded by the 5s and 5d shells. In addition, f-f transitions are forbidden, providing sharp, narrow, and iconic emission peaks. Lanthanide ions have a narrow absorption cross section and limited absorption efficiency, making it difficult to directly emit light through photoexcitation. This problem is solved by the “antenna effect”, which mainly realizes the luminescence of lanthanide–organic complexes by transferring energy to lanthanide ions through organic ligands. The “antenna effect” refers to the formation of Ln–organic complexes through coordination bonds between lanthanide ions and organic ligand molecules with a high light-absorption coefficient [35]. With the help of the large absorption of the organic ligand in the ultraviolet region, the characteristic emission of lanthanide ions is greatly improved, and its energy is transferred to the emission level of lanthanide ions through an efficient intramolecular energy-transfer process.

As shown in Figure 1, the specific process of energy absorption and conversion of lanthanide–organic complexes is embodied in three steps: First, the ground-state organic-ligand chrominance group in the lanthanide coordination polymer absorbs the excitation light and is excited from the ground state (S_0_) to the first excited singlet state (S_1_), and the excited electrons return to the ground state by radiation or non-radiation means, or the first singlet state (S_1_) can shift the system to the triplet excited state (T_1_). Finally, the T_1_ state of the ligand is transferred to the metal ion, resulting in the highly sensitive luminescence of the lanthanide ion. Subsequently, the energy of the lanthanide ion decreases from the excited state to the ground state, and the characteristic fluorescence of the lanthanide ion is emitted. However, the energy transfer from ligands to lanthanide ions is incomplete, and the intramolecular transfer of ligands exists in the form of radiation (π-π*, n-π*). Most lanthanide coordination polymers exhibit the characteristic emission colors of lanthanide metal ions, such as Tb^3+^, Tm^3+^, Eu^3+^ and Sm^3+^, which are green, blue, red and orange, respectively, while Nd^3+^, Yb^3+^ and Er^3+^ emit in the near infrared region [36].

The diversity of organic ligands makes Ln–organic complexes exhibit different responses to different fluorescence-sensing analytes, and, hence, Ln–organic complexes are expected to be efficient and multifunctional fluorescence-sensing materials [37,38,39]. As a fluorescence-sensing material, Ln–organic complexes indicate some potential advantages: (1) the diversity of functional ligands with different molecular sizes ensures that the prepared Ln–organic complexes are suitable for the requirements and goals of the detection of analytes; (2) the pore characteristics of Ln–organic complexes are conducive to the adsorption of analytes and the increase in analyte concentration, making them closer to the host–guest interaction sites and decreasing the low detection limit of the analytes; and (3) the high specific surface area of Ln–organic complexes can enrich the analyte and improve the sensitivity of detection. Moreover, sensors prepared with Ln–organic complexes also have special advantages: (1) compared with high-performance liquid chromatography and atomic absorption spectrometry, they are easy and convenient to operate and low-cost, and there is no special requirement for the instruments; (2) the flexible porosity limits the distance between the analyte and the complex, ensuring the interaction between the complex and the analyte; and (3) the interaction site of the object is easy to identify, and the sensing mechanism is easy to infer and elucidate. Based on the abovementioned advantages, Ln–organic complexes are widely applied in the detection of metal cations, anions, nitro-aromatic compounds, antibiotics, pesticides, volatile organic compounds, biological small molecules, and biomarkers with high sensitivity and selectivity [40,41,42].

In this review, crystalline luminescent lanthanide–organic complexes with controllable structural features and excellent optical properties are discussed as zero-dimensional (0D) discrete structures, one-dimensional (1D) and two-dimensional (2D) coordination polymers, and three-dimensional (3D) frameworks. Many studies have reported that some nitroaromatics, such as nitrobenzene (NB), nitrophenol (2-NP or 4-NP), trinitrophenol (TNP), etc., can be detected by crystalline lanthanide–organic complex-based sensors. These explorations have led to the discovery of sensing materials with better detection performance. Moreover, various fluorescence detection mechanisms of lanthanide–organic complexes for nitroaromatics are summarized and sorted out to help other researchers better understand the fluorescence-detection mechanism of nitroaromatics and provide a theory for the rational design of new crystalline lanthanide–organic complex-based sensors.

## 2. Fluorescent Sensing of Nitrobenzene Pollutants Based on the Structural Characteristics of Ln–Organic Complexes

### 2.1. 0D Discrete Structure

The discrete-structured lanthanide complexes usually possess polynuclear metal secondary building units (SBUs) and the weak intramolecular interactions play an important role in the sensing of nitro-based pollutants. The reported discrete lanthanide complexes include Eu_2_ [43], Cd_8_Nd_4_ [44],Yb_18_ [45], Nd_42_ [46], and Yb_42_ [47] metal units, in which the polynuclear metal clusters exhibit characteristic nanoring structures (Figure 2). Ma et al. [44,45] prepared Nd_4_ and Yb_18_, and Shi et al. [46,47] synthesized Ln_42_ (Ln = Nd, La, Yb) nanowheel cluster structures. Due to their unique ring structures, special high-nuclearity lanthanide nanorings may show some advantages during the luminescent sensing, such as a strong capture ability for analytes. A great deal of intramolecular interactions existed in these lanthanide complexes, such as hydrogen bonds or Ar-H–π interactions, which resulted in the formation of high-dimensional supramolecular structures or three-dimensional channel structures, which are conducive to the interaction with analytes [44,45,46,47]. 

Discrete heterometallic d–f luminescent complexes are rarely synthesized due to harsh synthetic conditions. For instance, a series of Zn–Ln frameworks consisted of a long Schiff base ligand with a naphthyl backbone and two short polydentate ligands, in which the Zn–Sm complex showed visible and NIR luminescent sensing of nitro explosives and exhibited high sensitivity to 1,4-dinitrobenzene (1,4-DNB) and 2,4,6-trinitrotoluene (TNT) at the ppb level [48]. 

Investigations found that the discrete lanthanide complexes exhibited a strong lanthanide-characteristic visible or near-infrared (NIR) luminescent-sensing behavior for nitro contaminants with low detection limits in an aqueous solution or acetonitrile, such as nitrobenzene (NB), nitrophenol (2-NP or 4-NP) and 2,4,6-trinitrophenol (TNP). The selective detection of certain nitro contaminants can be attributed to several factors, such as the size of the channel of the Ln complexes or analyte molecules, and amount of nitro compounds. If the volume of the measured object is appropriate to enter the cavity of the sensor, the fluorescence intensity may change and, thereby, exhibit a significant sensing effect.

### 2.2. 1D Ln-Coordination Polymers

Several 1D-fluorescent lanthanide–organic complexes for sensing nitro pollutants were reported, possibly due to their monotonous structural characteristics and indistinct fluorescent sensing performance. One-dimensional Ln–organic coordination polymers [Ln(BDPO)(H_2_O)_4_] (Ln = Eu for CUST-623, Tb for CUST-624) can be prepared using the reaction of Eu or Tb ions with the N,N′-bis(3,5-dicarboxyphenyl)-oxalamide ligand (BDPO). [Ln(BDPO)(H_2_O)_4_] has a 1D framework structure with two kinds of 1D open channels in the *b*-axis direction. Such 1D Ln–organic complexes can be used as a fluorescent sensor to detect TNP with the low detection limit of 0.21 μM [49]. Further, 1D [Eu(L)_6_(DMF)] (L = 2-(2-formylphenoxy) acetic acid) can be packed into a 2D structure through hydrogen bonding, and it has a good selectivity as well as a high sensitivity to TNP with the detection limit of rapid response of 3.39 μM in a CH_3_CN solution (Figure 3). The good detection ability towards TNP over other nitro explosives may be ascribed to the possible quenching mechanism of competitive absorption, photoinduced electron transfer and hydrogen-bond interaction [50]. In addition, the mixed ligands are used to construct the 1D Ln–organic-coordination polymers, which is regarded as a good strategy. For instance, the reaction of Eu(III) ion with mixed ligands of 2,3,4,5-tetrafluorobenzoic acid and 1,10-phenanthroline afforded a 1D structural tetranuclear [Eu_4_] complex with two crystallographically independent Eu^3+^ ions, which exhibited a highly sensitive response toward nitrobenzene at the ppm concentration [51]. 

These 1D lanthanide complexes can be used as luminescence sensors to detect NB or TPA in an aqueous or CH_3_CN solution with the low detection limits at the μM level through the alteration in its luminescence property, exhibiting the characteristic visible luminescence of Eu or Tb ions. The fluorescence-detection mechanism of nitro contaminants involves photoinduced electron transfer (PET), resonance energy transfer (RET), competitive absorption of excitation energy (CA), and weak interactions. Moreover, a large number of hydrogen bonds in the 1D chain structure and weak interactions between complexes and analytes play an important role in the sensing process. 

### 2.3. 2D Ln-Coordination Polymers

Chemists explored a few 2D fluorescent lanthanide-based homometallic (f-f group) or heterometallic (d–f group) organic complexes as sensing materials for the detection of nitroaromatics. The studies revealed that the structural characteristics and the presence of weak interactions between 2D sheets (hydrogen bond, X…X bond, π stacking interaction) are beneficial to the interaction of the analyte with material and the fluorescence response of the luminescent species [52,53,54,55,56,57].

Liu et al. reported a new 2D Dy-organic complex [Dy(L)(NO_3_)(DMF)_3_]_n_ (H_2_L = 2,5-di(1H-1,2,4-triazol-1-yl) terephthalic acid, DMF = N, N-dimethylformamide)} with binuclear units as a ratiometric luminescent sensor for nitro compounds (such as 4-NP) in aqueous solutions; the lowest detection limit (0.0676 μM) indicates a high sensitivity [52]. The luminescence response of another 2D complex, [Eu(TFTA)_1.5_(H_2_O)_2_]·H_2_O (TFTA = tetrafluoroterephthalate), in methanol was influenced by the primary and secondary inner filter effects (IFE) of nitroaromatic compounds (4-NTP, 2,4-DNP and TNP). The modelling and correction of IFE revealed the mechanism of static and dynamic quenching in the complex, and F–F interactions were also involved in the assembly of 2D structures into 3D supramolecular entities; as predicted, these interactions weaken with the lanthanide contraction [53]. The similar Br–Br interactions were found in a series of 2D lanthanide coordination polymers {Eu_2_(TBrTA)_3_(H_2_O)_8_·2H_2_O}_n_ (TBrTA = tetrabromoterephthalate), the quenching rate of nitroaromatic compounds was 85%, including 4-nitrophenol, dinitrophenol, and trinitrophenol (picric acid), with the main quenching mechanisms of competitive absorption, photoinduced electron transfer, and the electrostatic process [54] (Figure 4).

Lanthanide coordination polymers constructed from mixed organic ligands exhibit rich structural features and good luminescent sensing properties for nitroarmoatics. However, lanthanide ions are not easy to coordinate with nitrogen atoms, and it is difficult to obtain mixed-ligand products by adding nitrogen-containing auxiliary ligands. It is through luck our group prepared a series of 2D lanthanide coordination polymers with a mixed ligand of 2/5-(pyridine-2-ylmethoxy) isophthalic acid and nitrogen-containing auxiliary ligands, which have high selectivity and sensitivity for the detection of TNP at nM level (Figure 5) [55]. Obviously, the addition of a nitrogen-containing ligand is conducive to the formation of non-covalent bonds and the promotion of the fluorescence sensing of nitrocompounds.

Although the preparation of 2D lanthanide-based heterometallic coordination polymers is a great challenge, a few complexes have been reported and showed fluorescent sensing properties for nitro contaminants. For example, f–f mixed-lanthanide complexes with different mixed ratios display a 2D sheet-like structure. Owing to the hydrogen bond interactions, the 2D structures are easily packed into 3D frameworks and exhibit a good sensing performance for 4-NP, such as 1-Eu_0.5_Tb_0.5_ [56]. In addition, 2D d-f-block Cd(II)-lanthanide(III) heterometallic-organic frameworks [CdCl(L)Eu_x_Tb_y_-(H_2_O)(DMA)](NO_3_)·3DMA (IFMC-36-Eu_x_Tb_y_) showed characteristic sharp emission bands of Eu(III) and Tb(III), and the intensities of red and green were tuned through changing the ratios of Eu(III) and Tb(III). It is worth noting that the luminescence property of IFMC-36-Eu suggests a potential application in the detection of the small-molecule pollutant nitrobenzene via significant fluorescence quenching [57]. 

### 2.4. 3D Ln–Organic Frameworks

Due to the high coordination number of lanthanide ions and their strong coordination ability with the oxygen atoms of the carboxylate ligand, the fluorescent lanthanide–organic framework with a 3D structure accounts for a large proportion of research. Owing to the advantages of the 3D microporous structure, the fluorescence-sensing performance of 3D lanthanide–organic frameworks is much better than those of 0D, 1D and 2D lanthanide–organic complexes; therefore, 3D lanthanide–organic frameworks are used not only for the detection of nitrobenzene, p-nitroaniline and 4-nitophenol, but also for 3,4-dinitrotoluene and 2,4,6-trinitrophenol.

The hydrothermal reaction of lanthanide ions (Ln^3+^) and 2,5-di(*1H*-1,2,4-triazol-1-yl)terephthalic acid (H_2_dttpa) formed lanthanide metal coordination polymers {[Ln(dttpa)_1.5_(H_2_O)_2_]·H_2_O}_n_ (Ln–CP, Ln = La^3+^, Ce^3+^, Nd^3+^, Sm^3+^, Eu^3+^) and {[Ln(dttpa)_1.5_(H_2_O)]·0.75H_2_O}_n_ (Ln = Tb^3+^, Er^3+^) [58], in which Eu–CP effectively sensitizes the visible emission of Tb^3+^ and shows high selectivity for Tb^3+^ and stable and high sensitive response with a minimal detection limit of 0.00988 μM; furthermore, Tb-CP acts as a good luminescence sensor to detect nitrobenzene (NB) with a detection limit of 0.0125 μM. Moreover, the fluorescence quenching of Tb-CP for NB can be attributed to the competitive absorption mechanism and the photo-induced electronic transfer mechanism of the excited-state interaction between the luminescent material and NB (Figure 6). 

In general, most studies were focused on Eu/Tb complexes for their excellent luminescence properties, but a few other Ln–organic complexes were also explored and applied to fluorescent sensors for the detection of nitroaromatic compounds. For example, the solvothermal reaction of La(NO_3_)_3_·6H_2_O, tetrakis(4-carboxyphenyl)ethylene (TCPE) and *L*-proline yielded the La metal–organic framework La(HTCPE)(H_2_O)_2_, which could be employed as a sensitive and selective fluorescence sensor for nitro-containing aromatic compounds, such as p-nitroaniline (NA). When NA was added into the suspension system of La(HTCPE)(H_2_O)_2_, fluorescence was almost completely quenched [59]. Due to the harsh synthesis conditions, 3D luminescent d-f block-based heterometallic organic complexes have rarely been synthesized; these deserve further exploration in the future to broaden the family of 3D luminescent Ln–organic complexes.

## 3. Detection Effect of Ln–Organic Complexes on Different Nitro Pollutants

Nitro pollutants are closely related to production and daily life due to their wide use in industrial processes, such as for pesticides, pharmaceuticals and dyes. It is of great significance to develop low-cost and effective fluorescence probe material for the detection of low-concentration hazardous nitro pollutants with high selectivity and sensitivity. 

### 3.1. Nitrobenzene (NB)

Nitrobenzene (NB), as the simplest nitroaromatic compound, is widely used in the synthesis of many products in industry, and leads to air, water, and soil pollution, as well as serious safety problems. Moreover, since NB is highly toxic, difficult to degrade, carcinogenic, and easy to settle, once it is exposed to groundwater and soil, it will have adverse effects on human health and the environment. The rapid and sensitive detection of NB is a crucial task; thus, many approaches have been developed for NB detection, such as chromatography, spectrophotometry, electrochemical method, and so on [37]. However, these methods are complex, expensive, and time-consuming. Many lanthanide complexes can be used as fluorescence materials to detect NB through the alteration in fluorescence properties and, usually, show high sensitivity and selectivity, due to NB inducing fluorescence quenching of lanthanide complexes in various organic solvents. As shown in Figure 7, a 3D Eu–organic complex {[Eu_2_(NSBPDC)_3_(H_2_O)_4_]·7(H_2_O)}_n_ (H_2_NSBPDC = 6-nitro-2,2′-sulfone-4,4′-dicarboxylic acid), as a fast and recyclable fluorescence sensor, exhibited good fluorescence sensing on NB via a turn-off response with a low detection limit of 11.315 μM [60].

However, there are few in-depth studies on the fluorescence quenching mechanism of lanthanide–organic complexes induced by adding NB. To investigate the mechanism of fluorescence quenching with NB, Liu et al. designed and prepared a Tb–FDA complex, {[Tb(FDA)_1.5_(DMF)]⋅DMF}_n_, using the reaction of Tb^3+^ and 2,5-furandicarboxylic acid (H_2_FDA) under solvothermal conditions, and used it as a highly selective and sensitive fluorescent probe for nitrobenzene and Fe^3+^. The fluorescence quenching mechanism was explained by the lowest unoccupied molecular orbital (LUMO) energy level of NB (−2.437 eV), which is lower than that of the H_2_FDA ligand (−1.954 eV) [61]. Hidalgo-Rosa et al. further expanded the knowledge of the selective-sensing mechanism of nitro compounds using luminescent terbium metal–organic frameworks [Tb(BTTA)_1.5_(H_2_O)_4.5_]_n_ (H_2_BTTA = 2,5-bis(1H-1,2,4-triazol-1-yl) terephthalic acid) through multiconfigurational ab-initio calculations [62]. 

### 3.2. Nitrophenol (4-NP or 2-NP)

Nitrophenol (4-NP or 2-NP) is one of the smaller nitroaromatics with one electron-withdrawing nitro group and one electron-donating hydroxyl group on a benzene ring. It is mainly used as intermediate of pesticides, medicines, dyes and other fine chemicals, and, therefore, causes serious pollution due to its high toxicity. The highly sensitive detection of NP in water is very important for water environment protection, human health and ecological environment safety. GC, AAS, MS, etc., are usually used for the detection of NP, but some potential disadvantages limit their applications [56]. Therefore, the use of efficient sensors for detecting NP is extremely convenient and significant; in particular, the development of Ln–organic complex-based fluorescence sensors with high sensitivity for NP is of pivotal practical significance. 

Lin et al. designed and prepared a series of {[Ln(HL)]·3DMF·3H_2_O}_n_ (Ln = Eu and Tb, H_4_L = 1,4-bis(2′,2″,6′,6″-tetracarboxy-1,4′:4,4″-pyridyl)benzene) (LZG-Eu and LZG-Tb)-based sensors to sense 4-NP with a detection limit of 0.0112 μM in deionized water, showing a promising Ln–organic complex sensor for 4-NP detection in actual water compared with those reported Ln complexes (Figure 8) [63]. 

To detect 2-NP, a few Ln–organic-complex-based sensors were also designed and prepared. For example, the large [Ln_42_L_14_(OH)_28_(OAc)_84_] (Ln_42_, Ln = Nd, Yb, HL = 3-methoxysalicylaldehyde) coordination polymer nanorings-based sensor exhibits NIR luminescence sensing for 2-NP in CH_3_CN with the detection limit from 8.22 to 27.1 μM [46,47]. The existence of the electron-withdrawing group on the benzene ring likely promotes the fluorescence quenching of lanthanide coordination polymers; thus, the sensing performance of Ln-CPs, for NP, has been discovered in recent years. 

### 3.3. 2,4,6-Trinitrophenol (TNP)

Picric acid (PA), or 2,4,6-Trinitrophenol (TNP), possesses three electron-withdrawing nitro groups and one electron-donating hydroxyl group on a benzene ring. As a common and typical malignant organic pollutant, TNP is used widely in industries for dyes, explosives, pharmaceuticals, fireworks, firecrackers and leather. High toxicity, and harmfulness to eyes, skin and the respiratory system inspired many researchers to efficiently detect TNP. The traditional detection methods, including HPLC, RS, IMS, GC, MS and MIT, could be used to detect explosives, such as TNP; some potential limitations of these technologies restrict their application [64]. TNP remains in a dissociated form in solution and can interact with the positive center on the fluorophore due to its low pKa value (pKa = 0.42) [35,65]; thus, the fluorescence sensing method is suitable for the detection of TNP in solution.

The Ln–organic complexes-based sensors can effectively detect TNP with high sensitivity and selectivity in aqueous solution. Xu et al. prepared a 3D [Tb(TCBA)(H_2_O)_2_]_2_·DMF (H_3_TCBA = tris(3′-carboxybiphenyl)amine; DMF = dimethylformamide) by the solvothermal reaction of Tb^3+^ with a propeller-like H_3_TCBA ligand. The resultant Tb–organic complex features 1D triangular channels. Such a structure indicated high selectivity, with an extremely low detection limit about 1.64 ppb implied a high sensitivity for TNP explosive [66]. The lowest detection limit means the highest sensitivity of the sensor, compared with the previously reported Ln-based sensors [39,44,45,49,50,53,55]. The strong host–guest interactions between the Tb metal–organic framework (Tb-MOF) and TNP are captured and accurately determined by online microcalorimetry, which provides a distinctive thermodynamic perspective to understand the heterogeneous sensing behaviors. Another interesting work about sensing TNP using an Ln–organic complex is that a proportional fluorescence probe RGH-Eu(BTC) for TNP recognition was prepared by fixing a rhodamine derivative (RGH) on the surface of the luminescent Eu coordination polymer Eu(BTC) [67]. In the presence of TNP, the red fluorescence emission of Eu(BTC) was quenched in the donor–acceptor electron transfer process. The green fluorescence emission of RGH is caused by a rhodamine spirallactam ring opening, and the RGH-Eu(BTC) color changes from colorless to orange, which is easy to observe with the naked eye. The sensor realizes highly selective sensing of TNP in acidic analytes (Figure 9).

Based on the abovementioned contents, fluorescent sensors based on Ln–organic complexes for nitroaromatics are listed in Table 1. 

## 4. Fluorescence Detection Mechanisms of Ln–Organic Complexes for Nitroaromatics

The fluorescence quenching mechanism of Ln–organic complexes by nitroaromatics with nitro electron-withdrawing properties may be caused by single or multiple reasons, which need to be further explored. Due to the presence of electron-withdrawing groups and the strong oxidant characteristic of nitroaromatics, when these compounds are exposed to electron-rich complexes, their unoccupied lower π* orbitals are capable of withdrawing electrons from the excited state of the Ln complex, resulting in fluorescence attenuation. The mechanism of detection of nitro pollutants by Ln–organic complexes can be divided into the following types: (1) resonance energy transfer (RET), (2) competition absorption (CA), (3) photoinduced electron transfer (PET); (4) structural transformation (ST); and (5) static or dynamic quenching [69].

### 4.1. Resonance Energy Transfer (RET)

Resonance energy transfer, as a photoluminescence-sensing mechanism, is a short-range non-radiative energy transfer process which can significantly improve quenching efficiency and sensitivity. When the excited donor (fluorophore) induces the acceptor (analyte) to fluoresce, the fluorescence resonance energy is transferred, while the fluorescence intensity of the donor is reduced. Energy transfer is a distance-dependent physical process, and the efficiency of energy transfer usually hinges on (i) spectral overlap extent, the emission spectrum and the absorption spectrum of the host and guest; and (ii) dipole–dipole interaction, and the distance and relative orientation of the host and guest [69]. The efficiency and rate of energy transfer depends mainly on the degree of spectral overlap between the donor’s emission spectrum and the acceptor absorption spectrum. If the energies of the donor and acceptor are close, the energy transfer is significantly likely to occur. The degree of spectral overlap can be determined by experiment. If the UV-vis absorption spectrum of the analyte overlaps with the emission spectrum of the complex to a certain extent, the resonance energy will transfer from the complex to the analyte, resulting in the fluorescence quenching of the complex. If the overlap between the emission spectrum of the complex and the absorption spectrum of the analyte is higher, the quenching efficiency is higher.

Both [Eu(BDPO)(H_2_O)_4_] (CUST-623) and [Tb(BDPO)(H_2_O)_4_] (CUST-624) based on N,N′-bis(3,5-dicarboxyphenyl)-oxalamide (BDPO) could be used as fluorescent sensors for detecting TNP, and the sensing mechanism could be attributed to RET because their fluorescence spectra overlap with the ultraviolet visible absorption spectra of TNP [49]. An Eu/Tb bifunctional metal–organic framework, [Eu_x_Tb_1−x_(L)(H_2_O)_3_]_n_ (H_4_LCl = 3-bis(3,5-dicarboxyphenyl)imidazolium chloride), was synthesized successfully through a solvothermal reaction and used as a luminescent sensor to detect 4-NP in DMF and the quenching efficiency was affected by RET (Figure 10) [66].

### 4.2. Competition Absorption (CA)

Competition absorption is the overlap between the excitation spectrum of an Ln–organic complex and the UV-vis absorption spectrum of the analyte. When the excitation spectrum of the complex overlaps greatly with the UV-vis absorption spectrum of the analyte, the complex and the analyte may competitively absorb the excitation light, thereby reducing the total available energy of the complex, resulting in a decrease in the excited state of the Ln–organic complex. As a result, the fluorescence property of the complex is quenched. For instance, a Dy organic complex [Dy(L)(NO_3_)(DMF)_3_]_n_ (H_2_L = 2,5-di(1H-1,2,4-triazol-1-yl) terephthalic acid) with multi-emission peaks can be used as a luminescent ratio sensor for 4-NP, and the sensing mechanism is attributed to a competitive absorption of excitation energies (Figure 11) [52].

### 4.3. Photoinduced Electron Transfer (PET)

The photoinduced electron transfer (PET) mechanism could be used to explain the luminescence quenching process of Ln–organic complexes caused by nitroaromatics. The luminescence of a lanthanide–organic complex is caused by the antenna effect, in which the energy is transferred from chromogenic organic ligands (sensors) to Ln^3+^ ions. For the PET mechanism, the conduction band (CB) energy levels of the chemosensors are higher than the LUMO levels of the explosives. Thus, the excitation energy of the chemosensors may be consumed by their electron transfer to the electron-deficient nitro explosives, resulting in the luminescence quenching of the lanthanide–organic complexes at the lowest LUMO energy value.

Various interactions in the metal coordination polymer (such as hydrogen bonding or π-π packing) with the host and guest of the analyte may facilitate the photoelectron transfer process during photoexcitation. PET is a redox process in which excited photoelectrons are transferred from the donor to the electron-deficient ground-state receptor. When the lowest unoccupied molecular orbital (LUMO) energy of the donor (fluorophores) is higher than the LUMO energy of the acceptor (analytical compound), photoelectrons may be transferred from the excited donor to the ground-state receptor, resulting in the fluorescence quenching of the donor. Correspondingly, when the LUMO orbital energy of the donor is lower than the LUMO orbital energy of the recipient, photoelectrons may be transferred from the recipient to the donor, resulting in the enhanced fluorescence of the donor. Then, a recombination of charges in the ground state of the donor or acceptor occurs. Therefore, the complexation between the analyte and the sensor probe material will cause changes in the highest occupied molecular orbital (HOMO) and LUMO of the probe, which will greatly affect the efficiency of the sensor probe.

Generally, the energy of the conduction band (CB) of an electron-rich MOF is higher than the lowest unoccupied orbital energy of a nitro compound. When the excited electrons of the CB of MOFs are transferred to the LUMO, the nitro compound will cause fluorescence quenching, and the fluorescence-quenching performance will be improved with the decrease in LUMO orbital energy. That is to say, the PET leads to fluorescence quenching. A lot of fluorescence-quenching examples of Ln–organic sensors for nitroaromatics are ascribed to the PET mechanism. For instance, {[Eu_2_(L2)_2_(H_2_O)_5_]·3H_2_O}_n_ constructed from 5-(3′,5′-dicarboxylphenyl) picolinic acid (L2) by an in-situ decarboxylation reaction under hydrothermal conditions was used as a multifunctional sensor to detect Cu^2+^ ion, MnO_4_^−^ anion, and NB. The results showed a high sensitivity and selectivity through the fluorescence-quenching effect (Figure 12). For the sensing of NB, a possible mechanism may be allied to the PET from an electron-rich excited MOF to the electron-deficient NB [68] (Table 1).

### 4.4. Structural Transformation Mechanism (ST)

The structural transformation mechanism is when guest molecules enter the pore skeleton of a metal–organic complex (host) to generate guest–host interactions, thereby leading to the alteration in the coordination environment of metal ions and even structural deformation, resulting in significant changes in the luminescence of the metal–organic complex. This case can be applied to the selective detection of analytes. The host–guest interaction promotes or interrupts the sensitization pathways of the lanthanide ion and, therefore, leads to the enhancement or quenching of the luminescence, respectively [70]. An investigation elucidated the detection principle of luminescence quenching in a [Tb(BTTA)_1.5_(H_2_O)_4.5_]_n_ sensor based on 2,5-bis(1H-1,2,4-triazol-1-yl) terephthalic acid (BTTA) for NB. Through multireference CASSCF/NEVPT2 calculations, it demonstrated the value of host–guest-interaction simulations and the rate constants of the radiative and nonradiative processes in understanding and elucidating the sensing mechanism in Ln-MOF sensors (Figure 13) [62].

### 4.5. Static or Dynamic Quenching Mechanism (SY)

Static quenching is the formation of a non-fluorescent ground-state complex between the complex and the analyte, which absorbs light but does not emit photons and immediately returns to the ground state. Dynamic quenching is the transfer of electrons between the quencher and the sensor through excited-state collisions. Both are generally distinguished by changes in fluorescence decay lifetime and two quenching processes. Fluorescence lifetime spectroscopy is a potential technique to distinguish between static quenching and dynamic quenching. The fluorescence lifetime spectra of static quenching before and after the introduction of an analyte are similar, and dynamic quenching decreases before and after the introduction of the analyte and, thereby, results in the fluorescence-quenching phenomenon of the complexes. Under the static mechanism, a non-fluorescent complex is formed between the complex and the analyte, and fluorescence derived from the same complex. However, the fluorescence lifetime was found to decrease in dynamic quenching, because the collisions between the complex and the analyte reduced the excited states. As shown in Figure 14, after adding TNP into a 2D Tb–organic network [Tb_2_(pyia)_3_(phen)_2_(H_2_O)]⋅H_2_O (H_2_pyia = 5-(pyridine-2-ylmethoxy) isophthalic acid, phen = 1,10-phenanthroline) solution, its fluorescence decay lifetime was 1.941 ns, reflecting a dynamic-quenching process compared with the average fluorescence lifetime value (1.402 ns) [55].

For many nitroaromatics, the fluorescence quenching of lanthanide compounds is mostly caused by a combination of various mechanisms. For example, the Nd_42_ cluster [Nd_42_L_14_(OH)_28_(OAc)_84_] (HL = 3-methoxysalicylaldehyde) exhibits interesting NIR luminescence-sensing behavior to nitro explosives, which may be explained by PET and RET mechanisms [46]. The Eu–organic complex {[Eu_2_(HL)_2_(H_2_O)_4_]·3H_2_O}_n_ (H_4_L = 3,3′,5,5′-azoxybenzenetetracarboxylic acid) can detect TNP with the high selectivity, which can be assigned to the combined effects of electron and energy transfer mechanisms as well as electrostatic interactions [71]. The emission of another Eu-MOF {Eu_2_(TBrTA)_3_ (H_2_O)_8_·2H_2_O}_n_ is more than 85% quenched by nitroaromatic compounds (NACs) such as 4-nitrophenol, dinitrophenol, and trinitrophenol (picric acid) with competitive absorption, photoinduced electron transfer, and electrostatic processes the main mechanisms of quenching [54].

## 5. Conclusions and Outlook

In summary, crystalline Ln–organic complex-based luminescent materials are elucidated and analyzed in this review. According to the different dimensions, lanthanide–organic complexes can be classified into 0D (discrete molecule), 1D, 2D and 3D structures. Correspondingly, only a few 0D, 1D and 2D Ln complexes with luminescent properties are found, and most luminescent Ln–organic complexes have the 3D structure. Due to their luminescent property, Ln-organic complex-based sensors can be used to detect nitroaromatics, including nitrobenzene (NB), nitrophenol (4-NP or 2-NP), and trinitrophenol (TNP) at very low concentrations. Moreover, six main fluorescence-detection mechanisms (RET, CA, PET, ST, SY) of Ln–organic complexes for nitroaromatics are addressed. This review will help readers to better understand the fluorescence sensing of lanthanide–organic complexes and their quenching mechanisms with nitroarenes.

In term of structural characteristics, lanthanide–organic-complex sensors possess, mainly, three-dimensional stable frameworks and large porosity. Investigations showed that low-dimensional lanthanide coordination polymers also exhibit good sensing performance due to their large specific surface area and the presence of large numbers of weak interactions. Therefore, it is necessary to explore more low-dimensional lanthanide coordination polymers as the sensing materials for the detection of nitroarenes with very low concentration. Due to the elusive synthetic conditions, lanthanide coordination polymer sensors with mixed ligands or heterometals are less reported. The rich structural features and better sensing properties of heterometallic Ln–organic complexes with mixed ligands motivate us to rationally design and prepare more such substances in the future for application in luminescence sensing. On the other hand, we hope that more and more nitroaromatics can be easily detected by Ln–organic complexes to increase the detection range and capability of nitro pollutants.

## Figures and Tables

**Figure 1 molecules-28-04481-f001:**
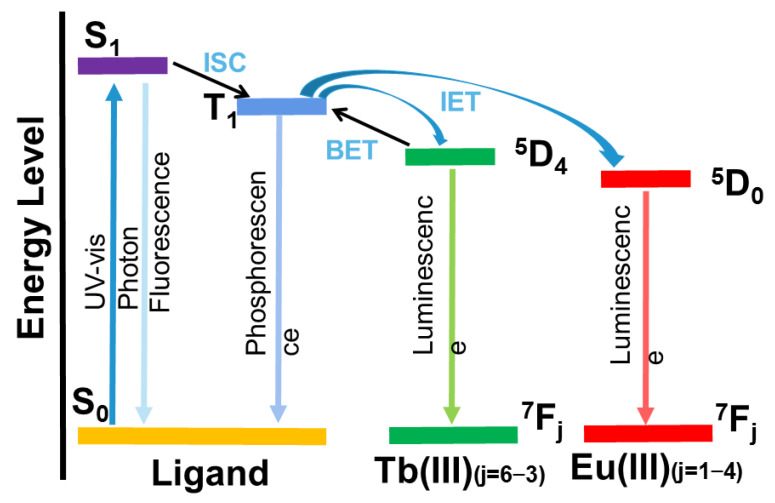
Process of energy absorption and conversion of Tb(III)/Eu(III) complexes. (ISC: intersystem crossing; IET: intramolecular energy transfer; BET: back energy transfer).

**Figure 2 molecules-28-04481-f002:**
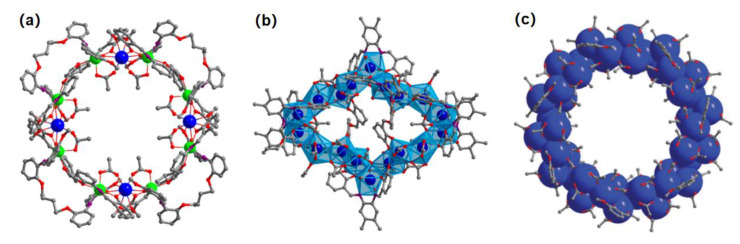
The high-nuclearity lanthanide nanorings. (**a**) Cd_8_Nd_4_ (Cd: green, Nd: blue, O: red, C: gray). Adapted with permission from Ref. [44], 2021, American Chemical Society. (**b**) Yb_18_ (Yb: blue, O: red, C: gray). Adapted with permission from Ref. [45], 2022, Elsevier. (**c**) Yb_42_ (Yb: blue, O: red, C: gray). Adapted with permission from Ref. [47], 2019, The Royal Society of Chemistry.

**Figure 3 molecules-28-04481-f003:**
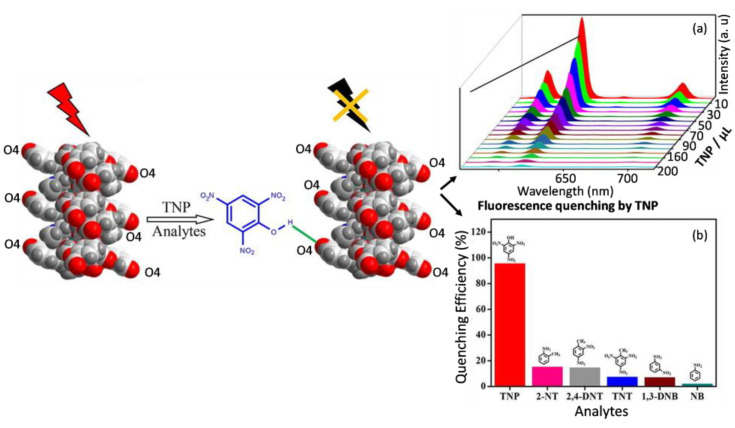
Schematic illustration of 1D Eu–organic complex [Eu(L)_6_(DMF)]_n_ (HL = 2-(2-formylphenoxy)acetic acid) having high selectivity and sensitivity to TNP in acetonitrile. (**a**) Concentration-dependent fluorescence quenching efficiencies of [Eu(L)_6_(DMF)]_n_ suspension based on the 618 nm peak. Gradually addition of TNP solution to the [Eu(L)_6_(DMF)]_n_ suspension (from 0 to 200 lL, kex = 350 nm); (**b**) The luminescence quenching efficiencies of different Nitro explosives towards the CH3CN dispersion of [Eu(L)_6_(DMF)]_n_. Adapted with permission from Ref. [50], 2022, Elsevier.

**Figure 4 molecules-28-04481-f004:**
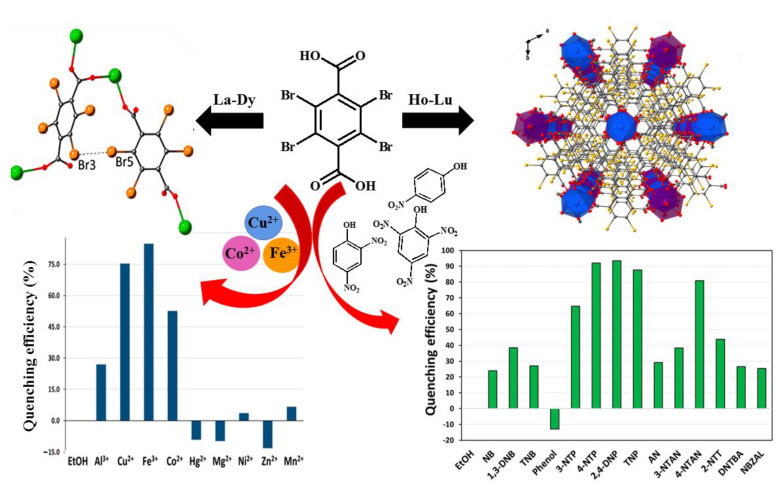
Scheme of design and detection for nitroaromatic compounds of {Eu_2_(TBrTA)_3_(H_2_O)_8_·2H_2_O}_n_ coordination polymer based on tetrabromoterephthalic acid. Adapted with permission from Ref. [54], 2018, American Chemical Society.

**Figure 5 molecules-28-04481-f005:**
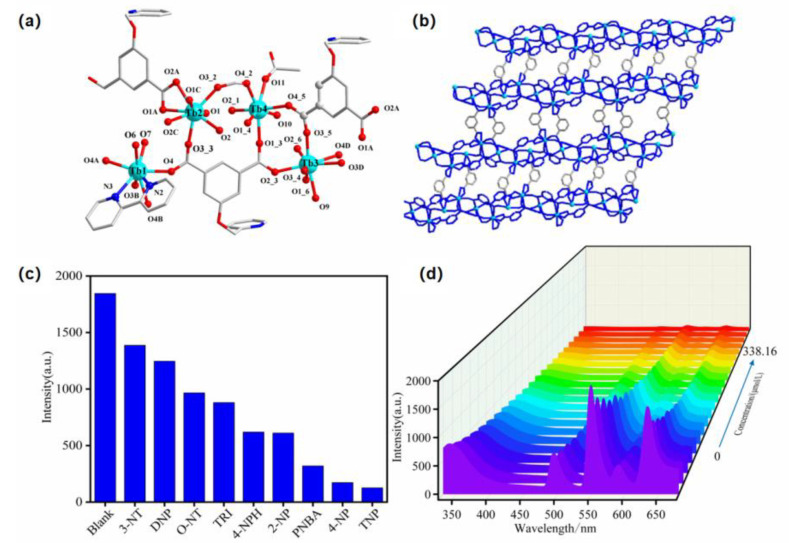
(**a**) The coordination environment of 2D Tb–organic complex based on mixed ligands (Tb: light blue, N: dark blue, O: red, C: gray); (**b**) 2D network; (**c**) the fluorescence intensity at 549 nm upon adding different nitro explosives (*λ*_ex_ = 313 nm); (**d**) changes in the fluorescence spectrum upon gradually adding TNP. Adapted with permission from Ref. [55], 2022, Elsevier.

**Figure 6 molecules-28-04481-f006:**
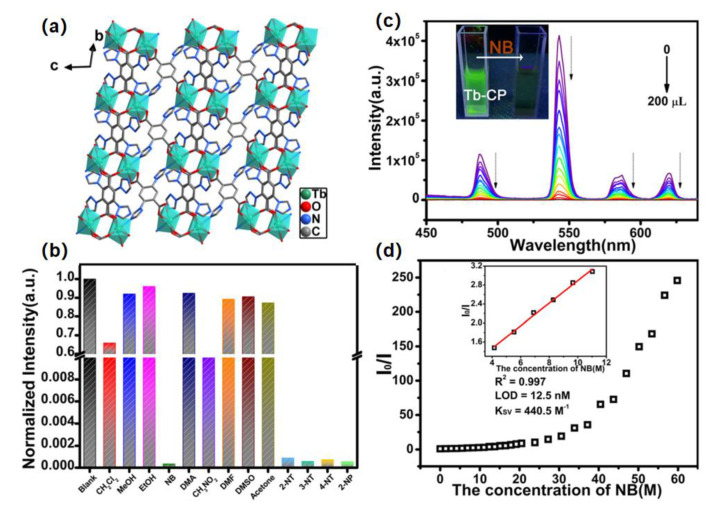
(**a**) Two-dimensional structure of {[Tb(dttpa)_1.5_(H_2_O)]·xH_2_O}_n_ (Tb–CP); (**b**) luminescence intensities of the ^5^D_4_ → ^7^F_5_ transition (542 nm) for Tb–CP CH_3_CN suspension with additional various solvents (0.1 mM) (λ_ex_ = 248 nm); (**c**) sensing for NB of Tb–CP; (**d**) curve plotting the luminescence intensity and concentration of NB (inset: Stern–Volmer plots of the linear part of the curve to obtain Ksv). Adapted with permission from Ref. [58], 2022, The Royal Society of Chemistry.

**Figure 7 molecules-28-04481-f007:**
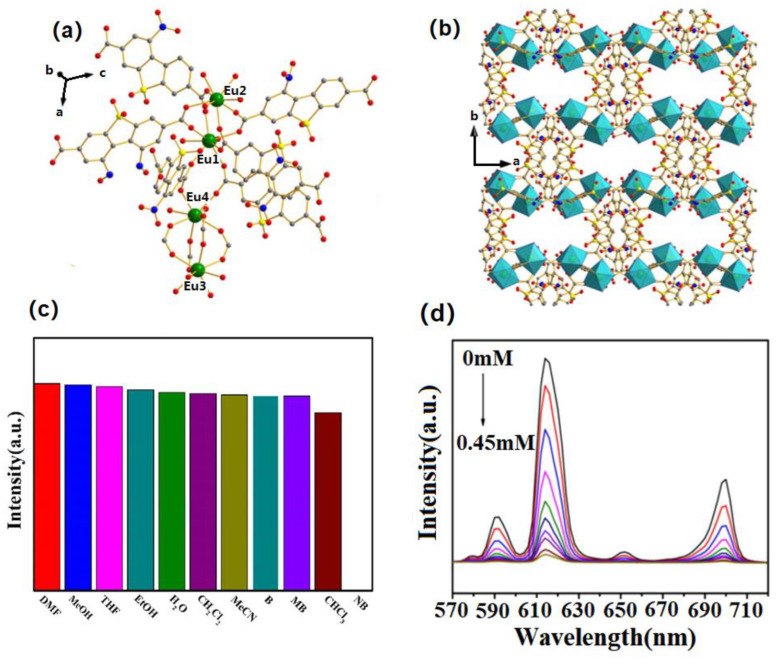
(**a**) Coordination environment of Eu^III^ ions in {[Eu_2_(NSBPDC)_3_(H_2_O)_4_]·7(H_2_O)}_n_ (Eu: green, N: blue, O: red, C: gray); (**b**) the 3D structure of {[Eu_2_(NSBPDC)_3_(H_2_O)_4_]·7(H_2_O)}_n_ with 1D channels along the a axis (Eu: green, N: blue, O: red, C: gray); (**c**) the luminescence intensity at 614 nm of {[Eu_2_(NSBPDC)_3_(H_2_O)_4_]·7(H_2_O)}_n_ dispersed in various solvents (λ_ex_ = 359 nm); and (**d**) luminescence intensity of {[Eu_2_(NSBPDC)_3_(H_2_O)_4_]·7(H_2_O)}_n_ upon gradual addition of NB. Adapted with permission from Ref. [60], 2019, Elsevier.

**Figure 8 molecules-28-04481-f008:**
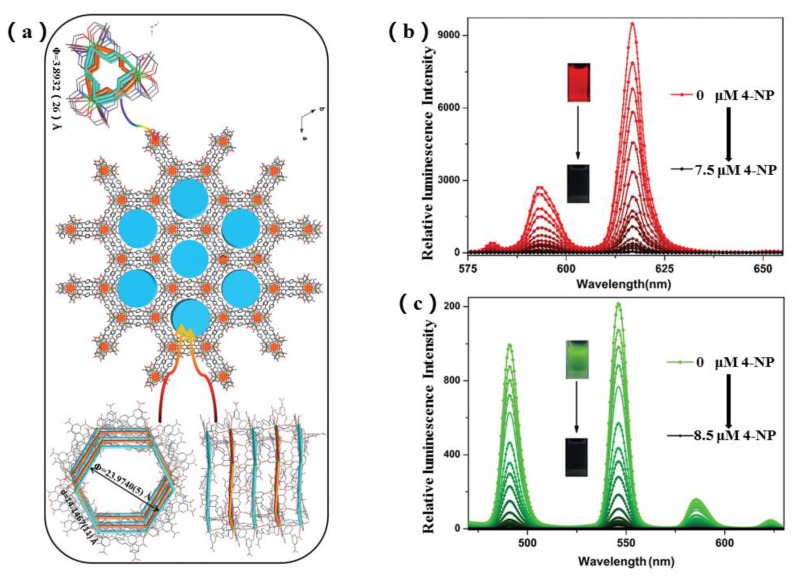
(**a**) The 3D molecular structure of {[Tb(HL)]·3DMF·3H_2_O}_n_ (LZG-Tb) showing the Tb_4_C_3_O_6_ subunit and chair conformational hexagon cavity viewing along c-axis (color modes: green, Tb; red, O; blue, N; gray, C); (**b**) the luminescence response of {[Eu(HL)]·3DMF·3H_2_O}_n_ (LZG-Eu) to different concentrations of 4-NP (0–7.5 μM) in deionized water; (**c**) the luminescence response of LZG-Tb to different concentrations of 4-NP (0–1 μM) in deionized water. Adapted with permission from Ref. [63], 2021, The Royal Society of Chemistry.

**Figure 9 molecules-28-04481-f009:**
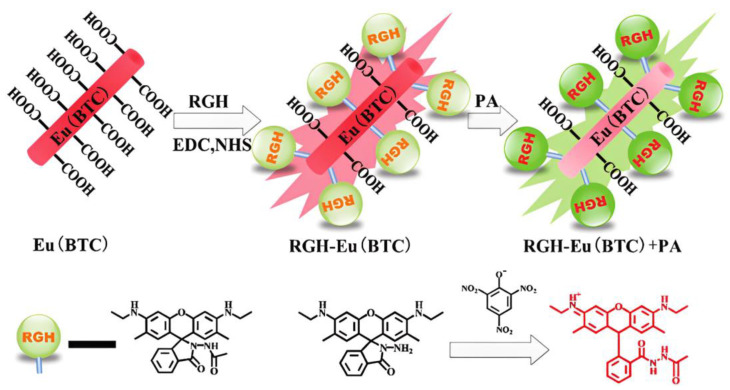
The synthetic protocol of RGH–Eu(BTC) and its principle for TNP (or PA) detection. Adapted with permission from Ref. [67], 2017, The Royal Society of Chemistry.

**Figure 10 molecules-28-04481-f010:**
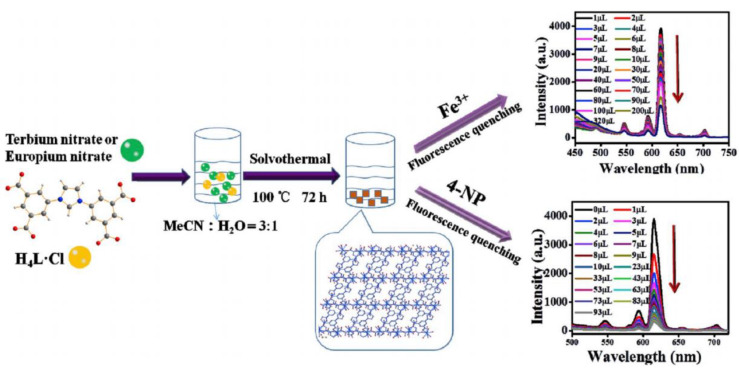
Schematic representation for the synthesis of complexes [Eu_x_Tb_1-x_(L)(H_2_O)_3_]_n_ and detection of Fe^3+^ cation and 4-NP through fluorescence quenching effect. Adapted with permission from Ref. [56], 2020, The Royal Society of Chemistry.

**Figure 11 molecules-28-04481-f011:**
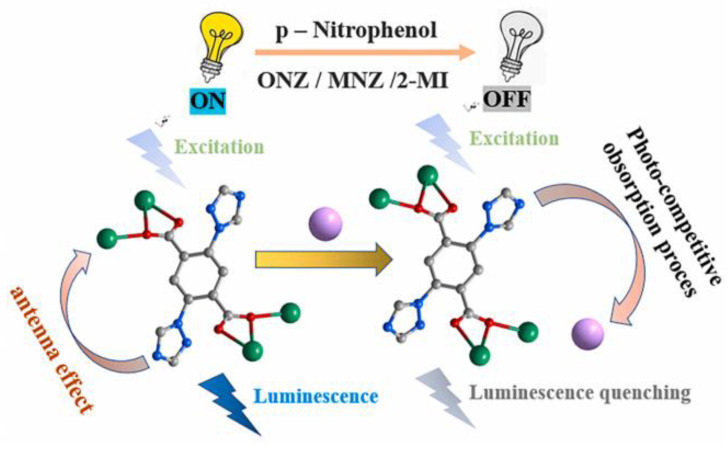
The schematic illustration of the mechanism of p-NP/ONZ/MNZ/2-MI sensing by [Dy(L)(NO_3_)(DMF)_3_]_n_. Adapted with permission from Ref. [52], 2021, Elsevier.

**Figure 12 molecules-28-04481-f012:**
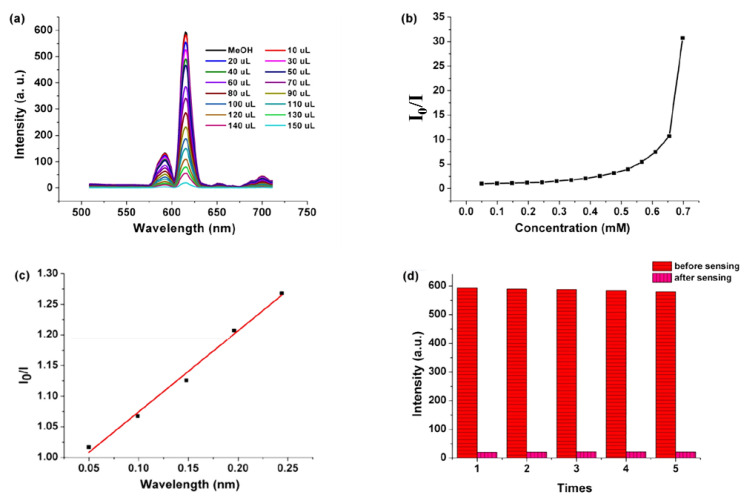
(**a**) Emission spectra of {[Eu_2_(L2)_2_(H_2_O)_5_]·3H_2_O}_n_ (Eu-MOF) in various concentrations of NB, (**b**) the Stern–Volmer plot of I_0_/I versus the concentration of nitrobenzene for Eu-MOF, (**c**) the S–V curve of I_0_/I versus the concentration of NB for Eu-MOF at low concentration, (**d**) the luminescence intensity (^5^D_0_ → ^7^F_2_) of five recyclable experiments of sensing for nitrobenzene in MeOH solution. Adapted with permission from Ref. [68], 2020, American Chemical Society.

**Figure 13 molecules-28-04481-f013:**
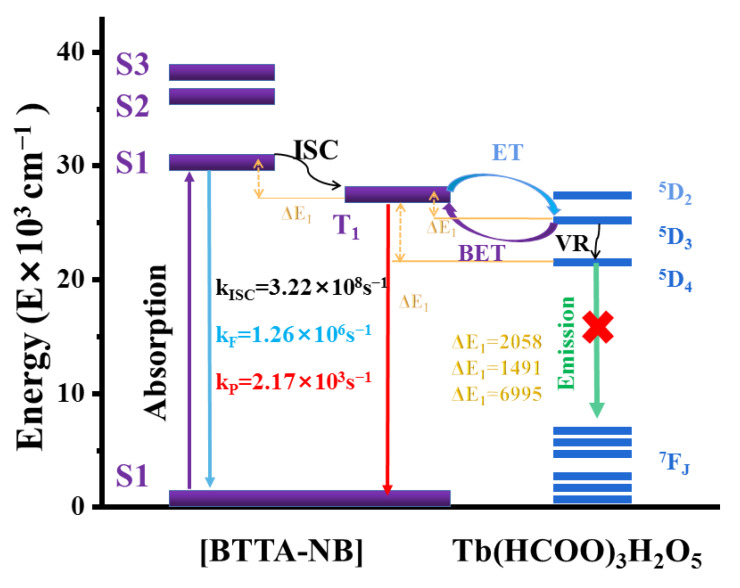
Energy level diagram depicting the most likely sensitization and emission processes for [Tb(BTTA)_1.5_(H_2_O)_4.5_]_n_/NB. Intersystem crossing, phosphorescence, and fluorescence rates are represented by the letters k_ISC_, k_P_, and k_F_, respectively, while ET stands for energy transfer and BET is back energy transfer (BET). Adapted with permission from Ref. [62], 2022, American Chemical Society.

**Figure 14 molecules-28-04481-f014:**
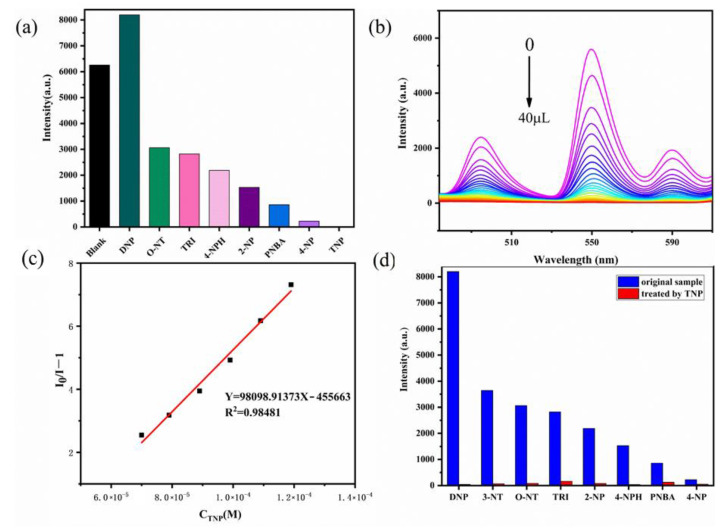
(**a**) Fluorescence intensity of Tb complex dispersed in aqueous solutions of different nitro explosives at 550 nm (λ_ex_ = 310 nm). (**b**) Tb complex dispersed in aqueous solutions of different concentrations of TNP at 550 nm; changes in fluorescence intensity. (**c**) S–V plot of TNP concentration and fluorescence intensity at low concentrations. (**d**) Interference of inorganic anions on Tb complex fluorescence sensing TNP. Adapted with permission from Ref. [55], 2022, Elsevier.

**Table 1 molecules-28-04481-t001:** Fluorescent sensors based on Ln–organic complexes for nitroaromatics.

Ln–Organic Complexes	Analytes	Detection Limit	Solvent	Mechanism	Refs
[EuL(H_2_O)_3_]·3H_2_O·0.75DMF	NB	none	DMF	Static	[38]
[Tb(FDA)(DMF)]·DMF	NB	0.01666 μg/mL	DMF	PET + Static	[41]
[Eu_2_(HPA)_6_(phen)_2_]·4H_2_O	NB	-	H_2_O	-	[43]
[Eu_2_(TFPht)_3_(phen)_2_(H_2_O)_2_]·H_2_O	NB	ppm	H_2_O	PET	[51]
[CdCl(L)Eu_x_(H_2_O)(DMA)](NO_3_)·3DMA	NB	-	H_2_O	PET	[57]
{[Ln(dttpa)_1.5_(H_2_O)]·xH_2_O}_n_	NB	12.5 μM	CH_3_CN	PET + CA	[58]
{[Ln_2_(NSBPDC)_3_(H_2_O)_4_]·x(H_2_O)}_n_	NB	11.315 μM	Ethanol	PET + CA	[60]
{[Tb(FDA)_1.5_(DMF)]·DMF}_n_	NB	-	DMF	PET	[61]
[Tb(BTTA)_1.5_(H_2_O)_4.5_]_n_	NB	-	Methanol	ST	[62]
{[Ln_2_(L_2_)_2_(H_2_O)_5_]·3H_2_O}_n_	NB	1.46 μM	Methanol	PET	[67]
[Tb_2_(TTHA)(H_2_O)_4_]	NB PNT TNP	0.489 0.609 0.919	Ethanol	PET	[42]
[Ln_2_(L)_2_(H_2_O)_2_]·5H_2_O	4-NP	7.6 × 10^−5^ M	H_2_O	CA	[40]
[Dy(L)(NO_3_)(DMF)_3_]	4-NP	0.0676 μM	H_2_O	CA	[52]
[Eu_0.5_Tb_0.5_(L)(H_2_O)_3_]_n_	4-NP	-	DMF	RET	[56]
LZG-Eu/Tb	4-NP	0.0112 μM	H_2_O	CA	[63]
{Eu_2_(TBrTA)_3_(H_2_O)_8_·2H_2_O}_n_	4-NP DNP TNP	17 μM 43.1 μM 74.6 μM	Ethanol	CA + PET + electrostatic	[54]
[Nd_42_L_14_(OH)_28_(OAc)_84_]	2-NP	8.22 μM	H_2_O		[46]
[Yb_42_L_14_(OH)_28_(OAc)_84_]	2-NP	27.1 μM	CH_3_CN	PET + other	[47]
[Tb_4_(2-pyia)_6_(HAc)_0.5_(2,2′-bipy) (H_2_O)_4.5_]·2,2′-bipy·H_2_O	TNP	0.0271 μM	H_2_O	CA + PET + dynamic	[39]
[Cd_8_Nd_4_L_8_(OAc)_8_]·4OH	TNP	0.55 μM	H_2_O	PET + inner filter effect	[44]
[Yb_18_(L^1^)_8_(HL[2])_2_(OAc)_20_]	TNP	5.1 μM	CH_3_CN	PET + other	[45]
[Tb(BDPO)(H_2_O)_4_]	TNP	0.20 μM	H_2_O	RET	[49]
[Eu(BDPO)(H_2_O)_4_]	TNP	0.21 μM	H_2_O	RET	[49]
[Eu(L)_6_(DMF)]	TNP	3.39 μM	CH_3_CN	PET + CA	[50]
[Tb_2_(pyia)_3_(phen)_2_(H_2_O)]·H_2_O	TNP	0.080 μM	H_2_O	CA	[55]
[Tb(TCBA)(H_2_O)_2_]·DMF	TNP	1.64 ppb	Ethanol	PET	[63]
RGH-Eu(BTC)	TNP	0.45 μM	Ethanol	PET + H-bonds	[65]
{[Eu_2_(HL)_2_(H_2_O)_4_]·3H_2_O}_n_	TNP	2.5 μM	H_2_O	RET + electrostatic interaction	[68]
[Eu(TFTA)_1.5_(H_2_O)_2_]·H_2_O	4-NTP 2,4-DNP 5-TNP	21.8 μM 31.2 μM 26.8 μM	Methanol	CA + inner filter effect	[53]
La(HTCPE)(H_2_O)_2_	NA	5.68 μM	DMF	PET + interactions	[59]
[Eu_2_(dtztp)(OH)_2_(DMF)(H_2_O)_2.5_]·2H_2_O	DCN	5.28 ppm	H_2_O	RET + PET	[32]

Note: NB: nitrobenzene; 2-NP: 2-nitrophenol; 4-NP: 4-nitrophenol; NA: nitroaniline; DNP: dinitrophenol; DCN: 2,6-dichloro-4-nitroaniline; TNP: 2,4,6-trinitrophenol; PNT: 4-nitrotoluene.

## Data Availability

No further data are available.

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
