# Peer review of "Advance Progress on Luminescent Sensing of Nitroaromatics by Crystalline Lanthanide–Organic Complexes"

_molecules, 2023, doi:10.3390/molecules28114481_

Round 1

Reviewer 1 Report

When I examine the entire review, I regret to say that the text has many inconsistencies. It was made sloppy in the text's writing and the presentation of the scientific data. Some are given below:

If the entire article is not edited, I think it should be rejected.

1. The resolution of the most figures is very low and not possible to read. Please supply the high resolution.

2. The authors should significantly revise the introduction, taking into account that in addition to the detection methods they have listed (p. 2, refs 16-18), there are many more methods for detecting nitroaromatic compounds. It should be especially noted that fluorescence detection methods can be used not only for the detection of nitrotoxicants in wastewater (according to the authors, see references 19-21). At the least, the authors should add modern reviews on the detection of nitrous-explosive compounds in the vapor phase (https://doi.org/10.1039/C9CS00778D; https://doi.org/10.1016/j.dyepig.2020.108414; https://doi.org/10.1016/j.fsisyn.2022.100298; https://doi.org/10.47191/ijmra/v6-i3-33; etc.)

3. The author should insert the structural formulas of the complexes when mentioning their abbreviations in the text (for example, on line 5 of СUST-623 and СUST-624). In addition, it is necessary to insert the structural formula of the complex in Figure 3.

4. Authors often do not refer to figures given in the article (for example, 7, 12, 13 and 14).

5. On page 10, the authors use the obscure term "electron-absorbing groups". What do they mean by this, referring to this type both the electron-withdrawing nitro groups and the electron-donating hydroxy group?

6. On page 10, the authors are mistaken in stating that the low detection limit of the sensor is indicative of its high selectivity, since these are different concepts. A low detection limit implies high sensitivity and nothing more.

7. On page 10, immediately after reference 57 is reference 62. Intermediate references are omitted.

8. References 58, 60, 61 are also omitted from Table 1. In addition, data on detection limits are given not for all complexes described in the review. Moreover, the authors should add a "solvent" column, since the detection limits is strongly dependent on the solvent in which the detection takes place.

9. Before each section concerning the detection of specific nitro compounds (Section 3), the authors do not refer in any way to the available methods for detecting these particular nitro analytes. For example, for the detection of nitrobenzene (Section 3.1.), a huge number of detection methods have recently appeared, both using fluorescent or electrochemical  methods and selective polymers with molecular imprints (see, for example, https://doi.org/10.3390/chemosensors9090255; https://doi.org/10.1016/j.foodchem.2021.131279; https://doi.org/10.1016/j.cjac.2022.100215; etc.). Authors are strongly encouraged to add appropriate references for each nitroanalyte.

10. Section "4.5. Chemical transformation mechanism (CT)" is not relevant and should be removed from the review, as only one reference is given and that one is for the detection of ascorbic acid (which is not a nitroaromatic analyte).

Moderate editing of English language.

Reviewer 2 Report

The manuscript presents an interesting and relevant overview of crystalline luminescent lanthanide-organic sensing materials for the detection of nitroaromatics.

In order to further strengthen the manuscript, I would like to suggest some revisions. Specifically, I kindly ask you to add more references to support the various sections of your manuscript, especially those discussing recent advancements and trends in the field of lanthanide-based sensors. This will ensure that your review provides a comprehensive and up-to-date understanding of the topic.

Additionally, I recommend providing more detail about the detection mechanisms, particularly in the section discussing resonance energy transfer (RET). It would be helpful if you could address factors that influence the quenching efficiency beyond the overlap between the emission spectrum of the complex and the absorption spectrum of the analyte. For example, please consider discussing how self-quenching of the complex and the analyte can impact the quenching efficiency. This additional detail will provide readers with a more thorough understanding of the factors governing the performance of lanthanide-based sensors for nitroaromatics detection.

I believe that incorporating these revisions will significantly enhance the quality and value of your manuscript.

Round 2

Reviewer 1 Report

The authors have made an efficient revision on their work, which can be accepted in the current form.

Minor editing of English language required.